# Fertility after Cancer: Risks and Successes

**DOI:** 10.3390/cancers14102500

**Published:** 2022-05-19

**Authors:** Chiara Di Tucci, Giulia Galati, Giulia Mattei, Alessandra Chinè, Alice Fracassi, Ludovico Muzii

**Affiliations:** Department of Obstetrics and Gynecology, “Sapienza” University, 00185 Rome, Italy; giulia.galati@uniroma1.it (G.G.); giulia.mattei@uniroma1.it (G.M.); alessandra.chine@uniroma1.it (A.C.); alice.fracassi@uniroma1.it (A.F.); ludovico.muzii@uniroma1.it (L.M.)

**Keywords:** infertility, gynecologic cancers, fertility sparing treatments, ovarian damage

## Abstract

**Simple Summary:**

Approximately one million new cases of cancer are diagnosed in women of reproductive age every year. In the last few decades, advances in early diagnosis and treatment have improved the survival rate. However, the adverse effects of anticancer therapy on the ovaries and uterus have a significant impact on future fertility and may affect the quality of life of cancer survivors. Unfortunately, evidence about the trend of ovarian reserve loss over time is insufficient for predicting the duration of the fertile period. Currently, impaired fertility in cancer survivors is a growing issue that is complicated by an increasing number of women delaying childbearing. This review focuses on the detrimental effects of chemotherapy, radiotherapy, and surgery on reproductive functions and describes the mechanisms causing reduced fertility in cancer survivors. Moreover, in this review, the available fertility preservation strategies to guarantee the chance of motherhood in cancer survivors are illustrated.

**Abstract:**

The incidence of cancer in reproductive-aged women is 7%, but, despite the increased number of cancer cases, advances in early diagnosis and treatment have raised the survival rate. Furthermore, in the last four decades, there has been a rising trend of delaying childbearing. There has been an increasing number of couples referred to Reproductive Medicine Centers for infertility problems after one partner has been treated for cancer. In these cases, the main cause of reduced fertility derives from treatments. In this review, we describe the effects and the risks of chemotherapy, radiotherapy, and surgery in women with cancer, and we will focus on available fertility preservation techniques and their efficacy in terms of success in pregnancy and live birth rates.

## 1. Introduction

The incidence of any type of cancer in 15–39 year old women is 52.3 rate per 100,000 [1].

In Italy, 3% of cancer cases are diagnosed in women under the age of 40. The most common types of cancer in women under 40 are breast cancer, thyroid cancer, melanoma, cervical cancer, endometrial cancer, ovarian cancer, leukemia/lymphomas, and colorectal cancer [2].

Breast cancer (BC) is the most common cancer in women under 49 (40% of all cancer cases). Despite the increased incidence of BC with age, approximately 7–10% of women diagnosed with BC are under 40 [3]. The incidence trend in Italy is slightly increasing (0.3% per year), whereas mortality continues to decline (−0.8% per year). The 5-year survival rate in young women (15–44 years) is 91% [4].

Thyroid cancer is common between the ages of 0–49 years (15% of all cancer cases). The most important prognostic factor is represented by the following histotype: the 20-year survival rate is 98–99% for papillary carcinomas and 80–90% for follicular ones (together they constitute 90% of thyroid cancer with papillary/follicular ratio 10:1), whereas it drops to 50–75% at 10 years for medullary carcinomas [5].

Melanoma is the third most frequent cancer in women aged between 18 and 39 years, with an increasing incidence trend (3.1% per year) [6].

Cervical cancer is the second for incidence in women. Its incidence increases with age up to 45 years with a peak between 45–55 years. The incidence rate is rising in developing countries but is decreasing in high resource countries [7].

Endometrial cancer has a very low incidence in reproductive age. Only 20% of endometrial cancers affect premenopausal women, and of these, no more than 5–8% affect women under 40 [8].

As for ovarian cancer, 80–90% occurs amongst women between the ages of 20 and 65 years old, and 90% of malignant tumors are diagnosed in women over the age of 40. A total of 5–10% of ovarian cancers are of intermediate malignancy (borderline) and approximately 30% of them affect women under 40. However, the diagnosis of epithelial ovarian cancer in women of reproductive age has become more frequent with increasing gynecological physical checkups [2].

Colorectal cancer’s (CRC) incidence is increasing in women of reproductive age. Fortunately, the 5-year survival rate from CRC is improving, with a survival rate of 65% [9].

Considering pediatric cancer patients, a recent study of over 3500 young women who survived childhood cancer such as leukemia, central nervous system cancer, Hodgkin lymphoma, non-Hodgkin lymphoma, Wilms tumor, neuroblastoma, soft tissue sarcoma, or bone tumor, shows a significantly higher risk of infertility than in the control group (RR: 1.48) [10,11]. There are still limited modalities available to preserve prepubertal fertility. Ovarian tissue reimplantation is the only fertility preservation technique that can be used to preserve prepubertal fertility, whereas mature oocyte cryopreservation is the main technique used for fertility preservation in post-menarche adolescents. However, a growing awareness and competence on the subject is beginning to emerge, especially amongst oncology pediatricians in Northern Europe [12,13].

In the United States, the 5-year overall survival rate for all invasive cancers between ages 15–39 is about 82.5% [14]. Despite the increased incidence of cancer cases, advances in early diagnosis and treatment have increased the survival rate [15].

Furthermore, in the last four decades, there has been a rising trend of delaying childbearing [16]. Hence, there is an increasing number of couples referred to Reproductive Medicine Centers for infertility problems after one partner has been treated for cancer. In these cases, the main cause of reduced fertility derives from the gonadotoxic effects of chemo/radiotherapy treatment [17]. As far as young women are concerned, there are two main concerns: the possible harmful effects of previous anticancer treatments on a future pregnancy, and the consequences pregnancy may have on the patient, particularly in the case of endocrine-sensitive neoplasms, even if to this day, there is no evidence of such possible adverse effects [18,19,20,21,22,23]. All reproductive-aged cancer patients must therefore be adequately informed of the risk of fertility loss/reduction as a consequence of anticancer treatments, and at the same time the strategies available to reduce this risk.

### Hematological Malignancies

Hematological malignancies are 7–9% of cancers in women. Acute lymphocytic leukemia, acute myeloid leukemia, non-Hodgkin’s, and Hodgkin’s lymphoma are the most common hematological cancers in reproductive age. Therapies used to treat these cancers are often gonadotoxic action. Gonadotoxicity and risk of premature ovarian failure (POI) depend on the type and stage of disease, the type and dose of anticancer therapy, and the patient’s age at the beginning of treatment. Cytogenetic and molecular abnormalities can influence the response to treatment, making additional treatments necessary in refractory or relapsing cases. With increasing overall survival rates in hematological cancers, preservation of fertility in these patients has gained increasing attention. [24,25,26,27,28] 

Validated options are freezing embryos and oocytes. In many cases, however, immediate cancer therapy is required, making it impossible to apply these options. Furthermore, ovarian stimulation is not possible in prepubertal age due to an inactive hypothalamus-pituitary-ovarian axis. Questionable options are: use of gonadotropin-releasing hormone analogues during chemotherapy, which, however, don’t protect the ovaries when TBI is used; surgical transposition of the ovary (oophoropexy) and gonadal shielding, which, however, does not protect ovaries when systemic chemotherapy is used.

Experimental options are: freezing ovarian tissues and following autotransplantation with potential contamination (especially in case of leukemia) of ovarian tissue with malignant haematological cells; in vitro maturation of oocytes in prepubertal age; an artificial ovary; stamina cells; neoadjuvant cytoprotective pharmacotherapy, and others [29,30,31,32,33,34,35,36]. Each of these options has advantages and disadvantages, so not all of them may be possible for all patients.

Preservation of fertility must be adapted to the patient [37,38,39,40,41]. A multidisciplinary approach between oncologists, gynecologists, reproductive biologists, and researchers is essential to ensure a high standard of care. Regarding fertility outcomes, ovarian stimulation followed by in vitro fertilization and embryo transfer is the most established strategy in postpubertal age. Pregnancy rates after frozen embryo transfer range from 36% to 61%, based on the patient’s age. Alternatively, oocyte freezing can be performed but randomized controlled trials show a pregnancy rate from 4.5 to 12% [42].

## 2. Ovarian Reserve in Female Cancer Survivors

Clinically, cancer treatments may compromise ovarian function, triggering delayed or arrested puberty, subfertility, infertility, and premature ovarian insufficiency (POI: menopause before the age of 40) [43,44].

Velez et al. first demonstrated, thanks to a population-based cohort study, that when comparing two groups of infertile women of the same age (34.8 years and 34.9 years, respectively), the infertility diagnosis was higher in cancer survivors (RR 1.30; 95% CI 1.23–1.37) compared to healthy women [45].

The reduction of ovarian reserve is a common cause of female infertility, especially among cancer survivors. 

Cancer treatments may alter ovarian reserve for different reasons [46]. The ovary is particularly sensitive to the side effects of chemotherapy and radiotherapy. The risk of ovarian failure depends on the agent used and the dose administered [47], and ovarian failure can happen during or after treatment [48].

Traditionally, the ovarian reserve was studied with Follicle Stimulating Hormone (FSH). However, FSH has several limitations, such as variability between cycles and during the same menstrual cycle. Recently, Anti-Mullerian Hormone (AMH) and ovarian antral follicle count (AFC) have been used as the main indicators of ovarian reserve. In particular, AMH is produced by the granulosa cells of preantral and antral follicles and inhibits recruitment of primordial follicles into the follicular pool [49].

Typically, AMH levels decrease during chemotherapy with slight recovery 3–6 months later. AMH, before and after treatment, can be helpful in the management of young women with cancer [50]. In women with breast cancer, AMH levels before the treatment have recently been shown to be a useful predictor of the post-chemotherapy loss of ovarian function, in addition to age, which is the only other established individual predictor [51,52]. 

Other studies have evaluated the impact of cancer therapy on the ovarian reserve of cancer survivors. In 2012, Gracia et al. compared ovarian reserve markers in a population of reproductive-aged cancer survivors and similar-aged controls. This study revealed a significant reduction of AMH and AFC in cancer survivors and a dose-dependent relationship between cancer therapies and markers of ovarian reserve. Furthermore, AMH and AFC levels were lower among cancer survivors than in the same age control group, although FSH values and menstrual cycles were regular, hence reflecting subclinical follicular depletion. Therefore, according to these authors, AMH and AFC seem to be the most sensitive markers to evaluate ovarian reserve depletion. It seems likely, but uncertain, that decline of those markers is a good reflection of fertility and/or predictor of time of menopause in cancer survivors) [53].

According to the study by Nielsen et al., female childhood cancer survivors may be at risk of infertility. The study compared 10-year cancer survivors to controls and found that patients treated with gonadotoxic drugs had statistically significantly lower AFC values than the control group (median 15 vs. 18, *p* = 0.047) On the other hand, the anti-Mullerian hormone values were not significantly lower (AMH) (median 13.0 vs. 17.8 pmol/L) [54].

In 2017, Wenners et al. studied breast cancer patients undergoing chemotherapy and confirmed the importance of AMH, AFC, and age as predictors of ovarian function. Additionally, they studied the effect of smoking, demonstrating its negative impact on ovarian activity and reserve [55].

In 2018 Van den Berg et al. evaluated, on a large scale, the long-term effects of childhood cancer treatment on both ultrasonographic and hormonal markers of ovarian reserve. Their findings showed that the ovarian reserve markers were remarkably low only in a minority of childhood cancer survivors (CCSs), even after treatment with alkylating CT (6.5–13.0%). However, the proportions of CCSs with abnormal ovarian reserve markers increased faster after age 35, compared to the controls. Hence, these women should be advised to try to conceive earlier in their reproductive lifespan, as it might prove shorter than anticipated [56].

Moreover, they found several discrepancies between combinations of ovarian reserve markers. In particular, low AMH levels were not associated with abnormal AFC, FSH, or inhibin B values, especially in young women; therefore, AMH may decline earlier than other markers in the sequence of events leading to menopause.

In conclusion, further studies are needed to confirm recent evidence regarding the reproductive potential of these patients.

Future studies should focus on the clinical value of each marker of ovarian reserve, in order to predict the chance of future pregnancy or to make a firm diagnosis of premature menopause in cancer survivors.

## 3. Influence of Cancer on Ovarian Function

If the cancer “per se” may induce alterations of ovarian functions, these should be investigated. 

Several studies have examined the fertility of women with cancer before chemo/radiotherapy and show mixed results. Different studies found a negative interference of cancer on ovarian function even before starting cancer treatment [57,58]. Pal et al. were the first that found an adverse effect of the tumor on ovarian function [59]. An observational study described how women with hormone-dependent cancer have a poorer response to controlled ovarian stimulation, with fewer oocytes retrieved than those with non-hormone-dependent cancer [60].

Alvarez and Ramanathan found significantly fewer oocytes in breast cancer patients than in patients with hematological cancer [61]. In a retrospective cohort study that included 155 women, Volodarsky-Perel et al. showed that women with grade G3 and stage III-IV breast cancer had significantly fewer numbers of mature oocytes than patients with grade G1-2 and at stage I-II of the disease. Also, the number of cryopreserved embryos was lower in women with grade G3. The same authors demonstrated that patients with high-grade cancer have fewer oocytes and embryos retrieved than those with low-grade cancer [62].

Similarly, Decanter et al. found fewer oocytes in women with cancer than in the age-matched control group [63]. Moria et al. showed that breast cancer patients had fewer retrieved oocytes than the control group [64].

However, most of the studies have not demonstrated relevant differences in ovarian reserve and oocytes retrieved between women with cancer and the healthy group. Almog et al., in their study with 81 women, stated that the ovarian function was not affected by the type of cancer [65,66,67,68,69].

These observations are thought to be linked to the systemic effect of cancer, causing higher catabolic status, increased stress hormones levels, and impaired function of the granulosa cells. Furthermore, invasive cancer infiltrates and destroys the surrounding tissue causing immune system reactions involving distant organs [70,71]. This systemic response conditions folliculogenesis, follicular cell proliferation, and oocyte maturation through the release of metalloproteases and growth factors [72,73].

Due to the close relationship between cancer and BRCA1 and 2 mutations, it is important to evaluate whether the cancer itself or the BRCA mutations underlie the reduced ovarian reserve in these women [74,75]. Porcu et al. demonstrated that BRCA 1 patients have a higher risk of premature ovarian failure compared to non-BRCA-mutated women with breast cancer and the healthy controls. BRCA1 groups have lower AMH levels and a significantly lower rate of mature oocytes. This effect seems to be independent of the probable interference of cancer. Therefore, the ovarian response after ovarian stimulation may be influenced by the presence of cancer [76].

More studies are needed to understand if there is a negative effect of cancer on ovarian function even before cancer treatment is started and if the type of cancer influences the ovarian response. 

## 4. Effects of Cancer Treatment on Female Reproductive Function

The adverse effects of cancer treatment on female reproductive function are an increasing problem that affects the quality of life in survivors of childhood, adolescent, and young adult cancer.

The three principal anti-cancer treatments are surgery, radiotherapy, and chemotherapy.

In most cases, primary ovarian or uterine cancers are surgically removed together with these organs, leading to sterility. For these women, using donated eggs or surrogacy are the only chance to have a child [77].

Chemotherapy and radiotherapy may damage the reproductive system by destroying the hypothalamic–pituitary axis, the uterus, or the primordial and growing follicles within the ovaries. 

The effects of chemotherapy and radiotherapy on the ovaries and uterus have a significant impact on the future fertility of childhood cancer patients as well as women up to the age of menopause.

### 4.1. Ovarian Damage

Cancer therapy can involve the administration of a wide variety of therapeutic protocols [78].

The American Society of Clinical Oncology published a useful classification of cancer treatments based on their level of gonadotoxicity and their consequent risk of permanent amenorrhea [79].

The level of gonadotoxicity depends on chemotherapeutic classes, dose, method of administration (oral versus intravenous), and combination of drugs; moreover, the toxicity changes with the type of disease, the woman’s age at the time of treatment, and the woman’s pre-treatment fertility. 

“Highly gonadotoxic treatments” cause permanent amenorrhea in at least an 80% of cases; the most common of them are, for example, chemotherapy used to treat breast cancer in women over 40 (combinations of cyclophosphamide, methotrexate, fluorouracil, doxorubicin, epirubicin), external beam radiation to a field that includes the ovaries, or myeloablative conditioning for hematopoietic stem cell transplantation with high-dose alkylating agents (cyclophosphamide/busulfan) in combination with total body irradiation [79,80]. Different studies have reported that the risk of gonadal insufficiency after stem cell transplantation is related to a woman’s age; the risk of infertility is 65–95% in adult women, higher than in prepubertal girls (around 50%) because of higher ovarian reserve [81,82].

Treatments classified as “intermediate gonadotoxicity” involve a 40–60% risk of amenorrhoea. These include adjuvant chemotherapy for breast cancer in women aged 30–39 (combinations of cyclophosphamide, methotrexate, fluorouracil, doxorubicin, epirubicin) and escalated (second-line) chemotherapy used for Hodgkin’s lymphoma [60].

“Low” gonadotoxic cancer treatments, with a risk of amenorrhea < 20%, include first-line treatment for Hodgkin’s lymphoma (ABVD therapy) and treatment for acute lymphoblastic and myeloid leukemia [77].

Treatments considered “very low risk” or “no risk” are antimetabolites (such as methotrexate, cytarabine) and vinca alkaloids (vincristine, vinblastine), which do not cause damage to human follicles [77,80].

#### 4.1.1. Impact of Chemotherapy

Many studies have investigated the type of damage of each chemotherapeutic agent on different cell types of the ovary. 

These findings have reported that ovarian damage can occur via several mechanisms [83].

Apoptotic death of primordial follicle and growing follicles are caused by direct DNA damage (via DNA double-strand breaks and inter-strand crosslinking) [80,84].

Direct damage to the ovarian stroma causes fibrosis and hyalinisation of small blood vessels, resulting in ischemia and necrosis due to a reduction in ovarian blood volume and consequently indirect damage to follicles growth [80,85].

Indirect damage to primordial follicles is due to increased follicle activation. Meirow’s group explained this mechanism of enhanced follicular demise owing to accelerated folliculogenesis by proposing the “burnout theory” [86,87]. In particular, the destruction of growing follicles and thus the local reduction in AMH concentrations causes an upregulation in the PI3K/PTEN/Akt signaling pathway. The direct consequence is that waves of activated primordial follicles were excessively recruited into growing follicles, which are more susceptible to the direct damage of chemotherapy, resulting in follicle burnout [80]. Table 1.

#### 4.1.2. Impact of Radiotherapy

Radiation therapy is one treatment modality for various types of malignancies, but unfortunately, exposure to ionizing radiation can lead to acute and long-term damage.

The effects of radiation to the abdominopelvic region depends on the dose intensity, fractionation, field of irradiation, and the age of the patient [77,88,89]. Women in the prepubertal period have ovaries relatively more resistant to gonadotoxicity [90].

The radiosensitivity of oocytes is high and differs according to their growth phase. In particular, dividing granular cells appear to be the main target of radiation-related gonadotoxicity. 

Radiotherapy-induced ovarian injury also involves the stroma with vascular damage, leading to tissue atrophy and fibrosis [91].

The underlying mechanism induced by radiotherapy is both direct and indirect. The radiation induces a direct ionization of the cellular macromolecules, such as gonadal DNA causing multiple lesions within the helical turns of the DNA, which is referred to as “direct” damage. Radiotherapy also leads to the generation of reactive oxygen species (ROS) in cells, increasing oxidative stress and diminishing antioxidant defense mechanisms. 

This imbalance may play a role in the etiology of radiotherapy-induced gonadotoxicity, which is defined as “indirect” damage [92].

Through these pathways, radiotherapy can affect healthy normal tissues in the ovaries and can influence the length of a women’s fertile lifespan and the timing of menopause.

### 4.2. Uterine Damage

Recent trials have reported that anti-cancer treatments can cause permanent injury to the uterus and compromise its ability to allow and sustain a healthy pregnancy [93].

#### 4.2.1. Impact of Chemotherapy

The percentage of pregnancies obtained through egg donation or using one’s own eggs before anticancer treatment is lower than in the control group. Even though patients underwent assisted reproductive technologies (ART), the implantation rate and clinical pregnancy rate (4.9% and 9.5%, respectively) were statistically significantly lower, underlining that cancer therapy-induced damage to the uterus may contribute to infertility [93,94].

A small number of studies have demonstrated that chemotherapy exposure during childhood (especially to alkylating agents) is associated with a smaller uterine size and volume [95,96].

Despite this indirect evidence of uterine damage induced by chemotherapy, there is a paucity of data regarding the specific pathological mechanism through which these drugs can act. Currently, the damage to endometrial epithelium, myometrium, uterine vasculature, and the endometrial stem cell niche can only be extrapolated from animal models or laboratory and clinical findings on the analogous cell line of other human tissue (for example intestinal stem cells, cardiomyocites, skeletal muscle) [93].

#### 4.2.2. Impact of Radiotherapy

Radiation to the uterus can impair reproductive function. Evidence has reported that radiotherapy can cause microvascular injury with endothelial damage and myometrial fibrosis compromising uterine growth and distensibility. Radiation may also damage muscle fibers and decrease pelvic floor muscle function [80]; when the woman’s exposure happens before puberty, stunted uterine growth and fibrosis can only be partially rescued by hormone replacement therapy [70]. Critchley et al. reported a shorter uterine length and a rare uterine blood flow in patients who received radiation during childhood compared to women with a history of POF (Primary Ovarian Failure) without radiation exposure.

The radiation consequences on the uterus are greater in the case of exposure at a young age, leading to greater risks for future pregnancy. The radiation dose that poses the greatest risk of reproductive failure is >45 Gy in adults or >25 Gy in childhood [80].

## 5. Oncofertility 

Approximately one million new cases of cancer are diagnosed in reproductive-aged women every year [93,97].

In the last few decades, life expectancy of these patients has increased thanks to anticancer treatments. The increased survival rate, combined with increased age for childbearing, has led to the occurrence of side effects such as fertility problems [98].

Diagnosis and treatment of tumors can often cause fertility problems in women. It has been estimated that 70–75% of cancer survivors are interested in parenthood and that 80% of them are affected by reduced fertility [99].

Despite patients’ interest in parenthood, the percentage of patients who receive correct information varies from 51% to 95%, and the percentage of patients who access fertility preservation techniques is low [100,101].

Fertility loss can cause devastating emotional reactions in women impacting their plans for the future. Various studies demonstrate that discussing fertility preservation options can improve quality of life and can contribute to psychological health [102,103]. 

The creation of an oncofertility team around the patient would allow this conversation to happen at the appropriate time and would reduce fertility loss in cancer patients [36]. 

Oncologists should inform patients about impaired fertility risk and should provide information on strategies available to preserve it.

Fertility preservation strategies in females depend on age, type of treatment planned, diagnosis, presence of a partner, time available before starting treatment, and potential for cancer to metastasize to the ovaries [79].

After consultation with a hematologist, oncologist, and specialist in reproductive medicine, an adequate consult can be carried out to evaluate ovarian reserve and gonadotoxicity of the therapies and to propose an appropriate fertility preservation technique [97].

### 5.1. Fertility Preservation Options

Different fertility preservation strategies can be proposed (Figure 1):Hormone Protection by Suppressing OvariesOophoropexyEmbryo Storage, Oocyte StorageOvarian Tissue StorageFertility Sparing Surgery

#### 5.1.1. GnRH-Analogues

In cancer patients who are candidates for chemotherapy, the use of GnRH analogues should be proposed but not considered as an alternative to cryopreservation [104]. 

GnRH analogues are used as chemoprotectors; used during chemotherapy they induce menopause, suppressing the ovarian cycle.

In 2018 and 2020, ASCO Guidelines and the European Society for Medical Oncology, respectively, recommended that GnRHa use should be offered to all cancer patients who desire to preserve fertility. [105]

In 2018, the British Fertility Society affirmed that GnRHa should be started immediately before chemotherapy and continued for the duration of therapy. [105]

GnRH analogues arrest ovarian cells in the G0 phase inducing cellular quiescence and making these cells less responsive to chemotherapy [106,107]; this treatment has shown effects in reducing primary ovarian insufficiency (POI) risk, increasing pregnancy rates, and having no negative effects on the cancer’s outcome.

The use of GnRH analogues can be also proposed in patients with hormone receptor-positive disease with safety [108]. 

Several studies have shown that the use of Goserelin has preserved fertility in a high percentage of patients affected by breast cancer [109,110].

The use of Goserelin reduces premature ovarian failure risk as well as prevalence of amenorrhoea and also improves disease-free survival and overall survival [111,112].

AMH can be used to evaluate the GnRHa protective effect on fertility [105]. 

#### 5.1.2. Oophoropexy

Ovarian transposition (oophoropexy) consists of surgical removal of the ovaries from the irradiation site.

This strategy can be proposed to patients who are candidates for pelvic radiotherapy (children and pre-menopausal women who desire to preserve fertility and prevent early menopause), in cases of gynecological or hematological cancers, such as cervical cancer and Hodgkin’s lymphomas, and in cases of medulloblastoma, urogenital rhabdomyosarcoma, pelvic sarcomas, Wilm’s tumor, and rectal cancer [107,113,114].

Before the procedure, a pelvic MRI should be performed to ensure that the tumor does not involve the ovarian region.

Ovarian transposition can be performed by laparoscopy or laparotomy (in case of concomitant resection of the tumor). One or both ovaries are relocated, either medially (behind the uterus in the case of Hodgkin’s Lymphoma), laterally, near the inguinal ring, in the paracolic gutters, or near the lower kidney pole (in the case of urogenital tumors, medulloblastoma, and rhabdomyosarcoma), or to any distant site.

At the end of the procedure, two metal clips should be applied to the transposed ovaries to make them visible on imaging.

Possible complications of oophoropexy are: torsion of the ovarian blood vessels, development of benign ovarian cysts, and subsequent chronic pain and ovarian and abdominal wall metastases at the trocar site [113,114,115].

Success rate of this treatment, in terms of preserved ovarian function, varies between 60% and 83% and several spontaneous pregnancies after this procedure are described. Nevertheless, some authors claim that these patients could require assisted reproductive technologies because of increased distance between the ovary and the fallopian tube. 

This increased distance compromises oocyte migration through the Fallopian tube and impairs fertility [114].

#### 5.1.3. Embryo and Oocyte Cryopreservation

Embryo cryopreservation is one of the options to preserve fertility. 

An oocyte and a sperm cell (obtained from a male partner or sperm donor) are needed to create an embryo [107]. 

Embryo cryopreservation is a good choice for patients with a stable relationship because of better pregnancy outcomes [116]. However, not all women have a stable relationship at the time of diagnosis; in this case, oocyte cryopreservation is a valid alternative, giving women an opportunity to procreate with a chosen partner in the future, without the need to fertilize the oocyte after retrieval [117,118]. 

Embryo and oocyte cryopreservation cause a two-week delay in chemotherapy initiation [35,36].

Ovaries are stimulated with daily injections of follicle-stimulating hormone and stimulation can be started at any point in the menstrual cycle.

Follicle growth is monitored by ultrasounds and blood tests (serum estradiol and progesterone levels). Ovulation is induced with an HCG injection when appropriate and oocytes are collected by transvaginal aspiration (with ultrasound guidance). The oocytes retrieved can be cryopreserved or fertilized in vitro to obtain embryos [79,119]. 

Oocyte and embryo cryopreservation, performed prior to anti-cancer therapies, are defined by the American Society of Clinical Oncology (ASCO) and the European Society for Medical Oncology (ESMO) as the most appropriate procedures to ensure motherhood in cancer survivors [120].

Among reproductive-aged women, breast cancer is one of the most frequent. In some cases, this is a hormone-sensitive cancer because of the expression of estrogen receptors.

In patients with these tumors, exposure to high estrogen levels can be risky. In order to avoid this exposure, oocytes can be recovered from a natural cycle or from ovarian stimulation protocol with aromatase inhibitor or tamoxifen (chemoprotective agents that have ovulation-inducing properties) [106,121]. 

#### 5.1.4. New Strategies

Among new strategies, immature oocytes retrieval is an experimental technique that involves cryopreservation of immature oocytes or matured in vitro. 

These oocytes can be used for vitrification or to obtain embryos by ICSI with partner sperm. This strategy allows for a reduction in the time needed for preservation and avoids exposure to hyperestrogenism caused by stimulation [122]. 

Mature oocytes can be obtained by in vitro maturation of immature oocytes (IVM) or in vitro activation of dormant follicles (IVA) [120]. 

Achieving pregnancy is possible using oocytes maturated in vitro [123] but pregnancy rates are lower in patients who have used embryos obtained from immature oocytes or oocytes matured in vitro than those who have used embryos obtained from mature oocytes [122]. 

After treatment, patients should be informed about their ovarian function using different parameters (AMH, FSH, estradiol) in order to decide whether to use cryopreserved oocytes or to start a new cycle [123]. 

The discovery of ovarian stem cells in the ovarian cortex, first found in mammals and then also in women, opened new chances to preserve fertility.

Despite these findings, in vitro maturation of ovarian stem cells (OSC)s to oocyte-like cells (OLCs) still needs to be investigated for future clinical use in female cancer survivors [120]. 

#### 5.1.5. Ovarian Tissue Cryopreservation

Ovarian tissue cryopreservation (OTC) is the only fertility preservation strategy available in prepubertal patients and those who cannot postpone treatment [35,124]. 

The entire ovary or part of this can be collected laparoscopically at any period of menstrual cycle. Obtained tissue is sliced and cryopreserved. 

When patients are declared free from cancer with a good prognosis, the ovarian tissue is thawed, tested to assess the absence of cancer cells, and reimplanted orthotopically or heterotopically [125].

Although the use of this technique has shown good results in adult patients, ex vivo maturation of ovarian tissue taken in childhood and the subsequent auto-transplantation is still considered experimental [126].

One of the problems of ovarian tissue transplantation is that revascularization occurs after a few days from the time of the procedure. This causes tissue ischemia and a loss of more than 60% of the primordial follicles [106].

Local administration of antiapoptotic and angiogenic factors can improve the revascularization of ovarian tissue [127].

Another problem with ovarian tissue transplantation is the possibility of transferring cancer cells in patients. Even though the tissue is controlled before freezing and before transplantation, the risk of tumor cell transmission remains.

This risk increases in certain tumors, leukemia being a case in point [128,129]. 

Therefore, this treatment is only proposed to patients with a low risk of ovarian metastasis [107]. 

Tumor cell transmission can be reduced by transferring primordial follicles onto an artificial tissue to replace native organs [130]. 

Depending on the number of follicles in cryopreserved tissue, ovarian function resumes after transplantation for 4–5 years on average and in some cases up to 7 years [118,131]. 

The first pregnancy after this fertility preservation technique was obtained in 2004; pregnancy and live birth rates are growing exponentially over the years. [35,132]. 

Patients who have undergone this fertility preservation strategy often have undetectable AMH levels, but spontaneous pregnancies after orthotopic transplantation have been reported [105].

AMH level is not associated with the duration of ovarian graft function or the possibility to achieve pregnancy in these women [133].

### 5.2. Success Rates

Cryopreservation of embryos, oocytes, ovarian tissue, or fertility preservation does not guarantee achieving a pregnancy in the future [36].

Among survivors, pregnancy rates are about 40% lower than the general population [16].

Several studies have described success rates after fertility preservation techniques (Table 2):Live birth rate (LBR) ranges from 20% to 45% in patients undergoing embryos-cryopreservation [134,135].Live birth rate varies from 20% to 50%, depending on age, in women undergoing oocytes cryopreservation [136,137,138].Live birth rate (LBR) ranges from 18.2% to 41.6% in patients undergoing ovarian tissue cryopreservation and reimplantation [132,137,139,140,141,142,143].Live birth rate (LBR) revolves around 8.9% in women undergoing in vitro maturation (IVM) [120].Live birth rate (LBR) revolves around 7% in patients undergoing in vitro activation (IVA) [120].Few studies have been carried out on pregnancy rates after oophoropexis. Despite this, pregnancies have been reported after this procedure by various authors [114,144].

## 6. Fertility Sparing Surgery for Gynecologic Cancer and Reproductive Outcomes 

All standard therapies for gynecologic cancer strongly impact a woman’s childbearing potential, with severe social and psychological effects. Preservation of fertility in some cases is possible and should be discussed during treatment planning, with reproductive-aged women with early-stage gynecologic cancer and with those who desire to preserve fertility.

### 6.1. Cervical Cancer

The standard surgical treatment for cervical cancer is a destructive and disabling surgery consisting of a radical hysterectomy and pelvic lymphadenectomy. However, in selected young patients, fertility preservation may be possible. In particular, it could be offered to patients with early-stage disease (tumor diameter ≤ 2 cm), negative nodes, and non-aggressive histological subtypes [145]. The CONTESSA/NEOCON-F multi-center study is intended to demonstrate the possibility of a conservative surgery to preserve fertility in patients with tumors diameter of 2–4 cm (FIGO 2018 stage IB2) with no lymphovascular disease after neoadjuvant chemotherapy [146] as well. According to preliminary data, live birth rates after ART in stage IB2 is only 9% [147]. Moreover, a retrospective study by Tesfai et al. evaluated the safety and the oncological and obstetric outcomes in patients with cervical tumors > 2 cm (FIGO stage IB-IIA) pre-treated with neoadjuvant chemotherapy before abdominal radical trachelectomy. In 15 out of the 19 patients treated (74%), fertility was successfully preserved and three of them became pregnant [148]. There are several fertility preservation techniques available: Dargent’s procedure; simple trachelectomy or cone resection; neoadjuvant chemotherapy with conservative surgery; and laparotomic, laparoscopic, and robot-assisted abdominal radical trachelectomy. In a recent systematic review, Bentivegna et al. examined these six fertility-sparing options to establish the best approach in terms of oncological outcomes and live birth success. All of these procedures showed optimum fertility results and were also dependent on the experience of the surgical team, but the best outcomes have been seen in patients with excellent prognostic factors, such as small tumor size, negative or little lymphovascular space involvement, and absence of positive margins [149]. These patients should be treated with conventional surgery alone, as their cases carry a minimum risk of recurrence.

#### Reproductive Outcomes

In a systematic review by the same group, the fertility rates and obstetric outcomes in a total of 2777 patients submitted to FSS (Fertility-Sparing Surgery) and 944 ensuing pregnancies were evaluated. The pregnancy and live birth rates were promising: 55% and 70%, respectively. The best approach in terms of fertility was found to be vaginal or minimally invasive radical trachelectomy. However, the live birth rate was similar in both invasive and non-invasive FSS. On the other hand, the prematurity rate was significantly lower in patients who had undergone a simple trachelectomy/cone resection rather than other conservative procedures [150]. In fact, obstetric complications seem to be higher in patients submitted to FSS. In a recent study, complications such as cervical stenosis and Asherman syndrome emerged as probable negative factors on fertility rate [151]. 

Nezhat et al. evaluated the reproductive outcomes in 3044 patients who underwent fertility-sparing surgery for early-stage cervical cancer (stage IA1–IB1), with encouraging results: the mean clinical pregnancy rate was 55.4%. The best surgical approach with the highest clinical pregnancy rate was vaginal radical trachelectomy (67.5%). The risk of recurrence and cancer death were very low (3.2% and 0.6%, respectively) after a median period of follow-up of 39.7% [152]. 

Schuurman et al. reported similar results: a vaginal approach of FSS in cervical cancer turned out to be the best procedure in terms of pregnancy and live birth rates. A possible explanation may be a less invasive resection of the cervix and parametrium [153].

Nevertheless, in a recent study, Tamauchi et al. investigated the response to controlled ovarian stimulation in patients after radical trachelectomy. 

They compared a group of 14 patients who underwent ovarian stimulation after radical trachelectomy with a control group who had male infertility or unexplained infertility. Estradiol levels and number of oocytes retrieved were lower in patients undergoing a trachelectomy than in the control group, despite the increased use of gonadotropin (3527.5 IU, SD 1313.4 vs. 2670.8 IU, SD 905.1, *p* = 0.001) [154]. 

In conclusion, fertility-sparing surgery must be considered a valid alternative to traditional radical hysterectomy for early-stage cervical cancer in women who desire fertility preservation. Adequate reproductive counseling before FSS is mandatory since many women are not conscious of the complications that could follow this kind of surgery and that could negatively impact the reproductive and pregnancy experience (Table 3).

### 6.2. Endometrial Cancer

Although rare, the incidence of endometrial cancer (EC) and atypical complex hyperplasia (ACH) is increasing in reproductive-aged women [155]. For both EC and ACH, the standard of treatment consists of a hysterectomy with bilateral salpingo-oophorectomy (ESGO/ESTRO/ESP guidelines for the management of patients with EC). However, the Society of Gynecologic Oncology’s Clinical Practice Committee describes as a perfect candidate for conservative treatment a patient who firmly desires fertility-sparing management with a well-differentiated (grade 1) tumor, stage FIGO IA without invasion of myometrium on MRI, absence of lymphovascular invasion, without intra-abdominal disease or adnexal mass, and no contraindications for medical management [156,157]. In these cases, according to the guidelines for the management of patients with EC, the options for conservative management are: progestogen (PG) therapy, LNG-IUD alone, or in combination with other hormonal agents [158]. The most successful fertility-sparing option is hormone therapy since ACH and EC, which express receptors for estrogen and progesterone, showed a higher chance of retaining fertility [159,160]. Medroxyprogesterone acetate (MPA) and Megestrol acetate (MA) are progestogen drugs commonly used in different doses (MPA 400–600 mg/day or MA 160–320 mg/day for a minimum of 6 months) for AEH and early-stage EC, but there is no consensus on the optimal treatment regimen. Nevertheless, in a recent systematic review, MA has been associated with higher rates of remission compared to MPA and other hormonal treatments [161]. Several studies have been conducted with the aim of estimating the oncological and reproductive outcomes of reproductive-aged women with EC or ACH. 

Wei et al., demonstrated that in these patients, conservative treatments can achieve good complete response rates (71% with PG and 76% with IUD alone); however, the pregnancy outcomes are worse in patients treated with IUD alone, rather than with progestin (18% vs. 34%) [162]. The same group reported a response rate for patients with LNG-IUD of 76%, and the recurrence rate was 9%. The pregnancy rate was 18%, and 14% for delivering a live newborn. When a combined therapy of oral PG plus LNG-IUD was used, then the complete response rate was 87%. [162].

However, a significant portion of patients experience recurrence after achieving complete remission. Traditional surgery is recommended in this case, but some patients who have not obtained pregnancy at the time of recurrence want to preserve their fertility. When recurrent diseases involve well-differentiated tumors confined to the endometrium, the second round of fertility-sparing therapy could be performed. A recent work conducted by Chen described fertility-preserving re-treatment that can have a promising response, with 88.6% of CR and a pregnancy rate after CR of 26.5% [163].

In the case of hysterectomy, or after radiation that will preserve the uterus but cause non-functionality in terms of future implantation and pregnancy, uterine transplantation should be offered to give women a chance to regain motherhood [164].

#### Reproductive Outcomes

Fertility outcomes in patients affected by endometrial cancer are not well described. A study by Park at al reviewed the medical history of 141 women with stage IA, grade 1 endometrioid adenocarcinoma who received progestin therapy. Seventy women (49.6%) tried to conceive, with a pregnancy rate of 73% (51 patients) and 58 live neonates [165]. On the other hand, Maggiore examined the fertility outcomes of levonorgestrel-releasing intra-uterine system in patients with EC or ACH. Among the 41 women who obtained a total surgical response, only 14 (34.14%) achieved a pregnancy (6 spontaneously and 8 through IVF) [166]. In a recent metanalysis by Harrison et al., a cohort of women aged ≤ 45 with EC or ACH was investigated. They received a standard treatment or a conservative treatment by progestin therapy. The live birth rate in these women was lower than that described in the literature, accounting for 11.6% of patients who received fertility-sparing management. In addition, among patients who achieved a pregnancy, 54% used some form of assisted fertility treatment [167] (Table 3). 

### 6.3. Ovarian Cancer

Approximately 3 to 13% of women with epithelial ovarian cancer are younger than 40 [168]. Premenopausal women with early-stage epithelial ovarian cancer may benefit from the feasibility and safety of FSS as they have a good cancer prognosis. [169]. In a systematic review, Bentivegna et al. described the conservative treatments that preserve reproductive function can be safe in women with unilateral epithelial cancers for stage IA grade 1 and grade 2, and IC grade 1, according to the FIGO staging system. Other patients with less favorable prognostic factors (bilateral ovarian involvement, any grade 3 disease, and stages IB, IC2, and IC3) must be informed that the safety of FSS is not confirmed. FSS is contraindicated for stage II/III (any histologic subtype) [170] anyway. The surgical approach consists of unilateral salpingo-oophorectomy with peritoneal washings and pelvic and para-aortic lymphadenectomy, as well as omentectomy. The risk of recurrence is about 8% and 10% in stages IA and IC. However, several retrospective studies confirmed that the overall survival in patients with early-stage epithelial ovarian cancer 10 years after diagnosis is up to 90% and found that survival was similar when comparing women who underwent fertility-sparing surgery with patients who underwent conventional surgery [171,172,173]. Borderline tumors represent a small but significant part of all primary ovarian tumors. It is possible to treat women with borderline tumors in clinical stage I with cystectomy or unilateral oophorectomy. The recurrence rates are 10–15% in a median 5-year follow-up, and the mortality rate is <1%. For women with bilateral borderline ovarian tumors, unilateral oophorectomy and contralateral cystectomy have been proposed as a reasonable technique in women with the desire to preserve fertility [174]. Ultraconservative management consisting of bilateral cystectomy has been proposed with excellent fertility results [175]. Non-epithelial cancers of the ovary represent approximately 5% of all ovarian malignancies and include ovarian germ cell tumors and ovarian sex cord-stromal tumors [176]. Malignant ovarian germ cell tumors (MOGCTs) are the most commonly found in women of reproductive age, and the FSS is generally the first choice of approach since these tumors are chemosensitive [177]. Experience to date indicates that the cure rates for women with early-stage MOGCTs approach 100% and at least 75% in advanced-stage disease [178].

Sex cord-stromal tumors generally have an indolent course and occur more likely in postmenopausal women. Frequently, if diagnosed in young women, these tumors are often seen as an early-stage disease and FSS is possible in a high percentage of young patients [179]. The oncological outcomes of tumors treated conservatively are reassuring, with similar results of conservative and radical surgery [180].

After fertility sparing treatment for non-epithelial cancer, there is a risk of POF. For this reason, it is necessary to accurately identify, through the Edinburgh selection criteria, the few girls and young women who will develop premature ovarian insufficiency, and who will be candidates for ovarian tissue cryopreservation (OTC). In the criteria, the conditions listed for OTC are as follows: a realistic chance of surviving for 5 years; a high risk of POI > 50%; and mild, non-gonadotoxic chemotherapy is acceptable if the patient is younger than 15 years of age [124].

#### Reproductive Outcomes

Any surgery that involves the ovaries may impact ovarian reserve and reduce the possibility of spontaneous conception. For these reasons, women who undergo fertility-sparing surgery may require the help of assisted reproductive technology, even if spontaneous conceptions have also been reported. Literature data regarding the pregnancy rate of women after FSS accounts for approximately 30% of all patients; however, if one includes only women of reproductive age, pregnancy rate increases and range from 66 to 100%, showing no relevant impairments after FSS [151]. A recent systematic review reported the reproductive outcomes of patients diagnosed with early-stage epithelial ovarian carcinoma (EOC), borderline ovarian tumors (BOT), or nonepithelial ovarian carcinoma (NEOC), are candidates for fertility-sparing surgery or conventional surgery. The fertility rates vary depending on the tumor type and were 76 to 96% for EOC, 65 to 95% for NEOC, and 23 to 100% BOT. The authors confirmed the safety of FSS procedures since the 5-year overall survival rates were almost overlapped between patients who underwent FSS (87.5%) and those who had conventional surgery (91.8%). 

Data in the literature regarding reproductive outcomes after MOGCTs are reassuring, between 66.6% and 95%. Regarding the pregnancy rate in the case of juvenile granulosa cell tumors, the data is encouraging, compared to adult granulosa cells tumors (42.3% vs. 28.5%) [180,181]. In fact, although preserving reproductive function is a goal of FSS, it has been reported that one in four women want to conceive after surgery [182]. These data require careful advice for women who decide to undergo FSS procedures rather than traditional ones, because while exposing themselves to an increased risk of cancer, these treatments do not necessarily guarantee future pregnancy for women. (Table 3).

### 6.4. Vulvar Cancer

The risk of vulvar cancer increases with a women’s age and it is 10 times higher in patients over 75 years of age [183]. Less than 20% of cases are in women younger than age 50 [156]. FSS should include less destructive surgery, and oophoropexy could be proposed to patients who are candidates for pelvic radiotherapy and OTC before chemotherapy, if possible. Chemotherapy can be associated with a lower risk of local long-term complications due to reduced local tissue damage [183]. No literature data are available about fertility outcomes in young women affected by vulvar cancer.

## 7. Special Categories

### 7.1. Fertility Preservation Post Cancer Diagnosis in Young Women

After cancer diagnosis, there is no suitable period to get pregnant. 

Completion of treatments, age of the patient, risk of relapse, ovarian function, and patient’s desire should be considered to determine the most appropriate time to conceive [16].

Discussion on patient fertility should be held both at the time of cancer diagnosis and at the end of treatments. 

AMH should be used to evaluate chemotherapy-associated ovarian dysfunction after treatments, but it is well known that AMH does not predict the probability to achieve pregnancy [105,184].

In the event of a reduced ovarian reserve (DOR), women should be advised that a low ovarian reserve does not affect the spontaneous conception rate if they wish to seek a short-term pregnancy. The oncologist should only reassess the safety of conception, with the patient, if the appropriate conditions are met and discuss the possibility of undergoing assisted reproductive technology (ART) procedures in case of infertility [36]. 

Women who presented cancer at a young age and who survived hide their fears about their reproductive health. Counseling about post-cancer preservation should be proposed and should not be seen as an alternative to fertility preservation at the time of cancer diagnosis, but as an additional option. 

Lehman et al. described fertility preservation in women after cancer with preserved ovarian reserve, diminished ovarian reserve (DOR), and POI [185]. Eleven subjects with DOR (52%) stored their eggs. Filippi et al. described oocyte cryopreservation in female cancer survivors, aged >18 years, who were diagnosed with DOR. Oocyte cryostorage was performed in 9 (69%) women [186].

### 7.2. Fertility Preservation in Women with Hereditary Cancer Syndromes

Hereditary cancer syndromes (HCS) are a group of genetic conditions associated with an increased risk of developing cancer during lifespan. Fertility preservation to prevent the effects of cancer treatments on ovarian function should be preceded by personalized and early fertility counseling. Carriers should be informed about oocytes cryopreservation, either before or after tumor diagnosis, but should also be aware of the risk of transmission [187]. There are no specific guidelines about fertility preservation in HCS, and only few papers, mostly focusing on BRCA-1/2 carriers, are reported. The ASRM guidelines give few recommendations for BRCA carriers. They recommend informing carriers of the possibility to undergo prenatal diagnosis in the case of natural conception or preimplantation genetic testing (PGT) and oocytes donation in the case of IVF [188].

## 8. Conclusions

Counseling of cancer diagnosis and treatment should be associated with adequate counseling about fertility preservation before treatment initiation, but also after cancer treatments. Women with cancer, regardless of their age, should be educated on the risk of endocrine dysfunction, cessation of ovarian activity, and depletion of the ovarian reserve. Furthermore, they should be assigned to fertility specialists to decide on fertility preservation options. In postpubertal women, where the beginning of chemotherapy can be postponed by at least 10–12 days, oocyte cryopreservation should be offered. OTC should be proposed instead when immediate chemotherapy is necessary both in prepubertal and postpubertal women. Evidence of the trend over time of ovarian reserve loss is insufficient to predict the duration of the fertile period. Therefore, it is important to inform women about the post-cancer depletion of the ovarian reserve in an attempt to induce them to modify their reproductive plans. If they have a partner, motherhood could be anticipated. In cases of single women not seeking pregnancy in the near future, one should consider the option of social freezing, as incidence of pregnancy decreases significantly over the years amongst cancer-treated patients. 

## Figures and Tables

**Figure 1 cancers-14-02500-f001:**
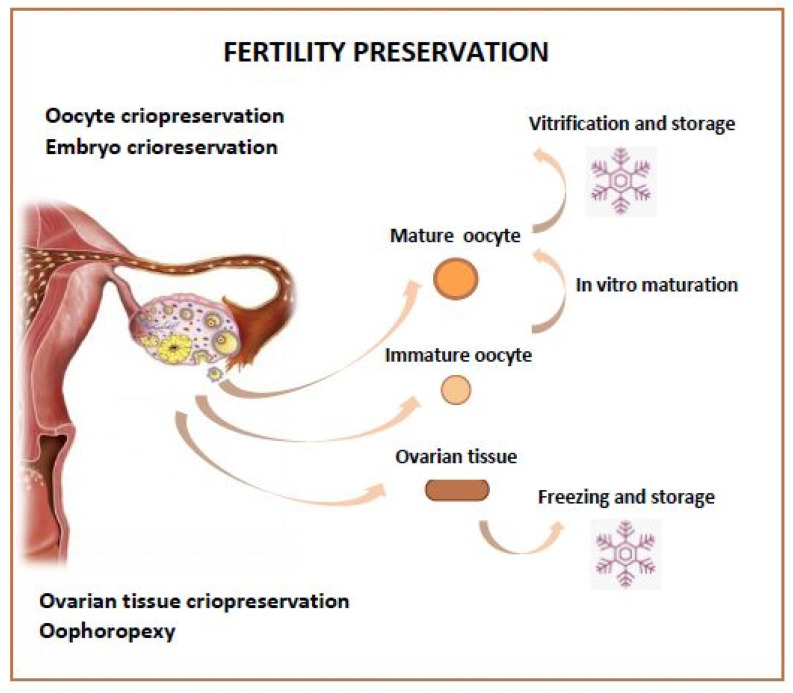
Fertility Preservation Options.

**Table 1 cancers-14-02500-t001:** Ovarian damage risk with chemotherapeutic agents.

Damage Risk	Cancer Treatment
High risk	CMF
	Cyclophosphamide
	Melphalan
	Busulfan
Intermediate risk	Cisplatin
	Carboplatin
	Oxaliplatin
	Doxorubicin
Low/very low	CHOP
	Vinblastine
	Vincristine
	Methotrexate
	5-fluorouracil

Notes: CHOP, cyclophosphamide/doxorubicin/vincristine/prednisone; CMF, ciclofosfamide, methotrexate e 5-fluorouracile.

**Table 2 cancers-14-02500-t002:** Live birth rate after fertility preservation techniques.

Techniques	Authors	LBR
Embryos-cryopreservation	Dolmans et al. 2015	20–40%
	Oktay et al. 2015	
Oocytes cryopreservation	Cobo et al. 2016	20–50%
	Diaz-Garcia et al. 2018	
	Specchia et al. 2019	
Ovarian tissue cryopreservation and reimplantation	Donnez et al. 2015	18.2–41.6%
	Van der Ven et al. 2016	
	Diaz-Garcia et al. 2018	
	Meirow et al. 2016	
	Donnez et al. 2017	
	Jensen et al. 2017	
	Shapira et al. 2020	
In vitro maturation	Silvestris et al. 2020	8.9%
In vitro activation	Silvestris et al. 2020	7%
Oophoropexis	Terenziani et al. 2009	Rare
	Irten et al. 2013	

**Table 3 cancers-14-02500-t003:** Reproductive and oncological outcomes of patients after undergoing fertility-sparing surgical procedure.

Type of Cancer	Author	Treatment	Patients		Reproductive Outcomes			Oncological Outcomes	
			Total	TTC (%)	N° Pregnancies	ART Pregnancies	LBR (%)	Recurrence (%)	Death (%)
Cervical cancer	Bentivegna et al. 2016	ST	212	NA	103	NA	74.0	NA	NA
		VRT	1355	NA	499	NA	67.0	NA	NA
		RT (lptm)	735	NA	175	NA	68.0	NA	NA
		RT(MIS)	314	NA	74	NA	78.0	NA	NA
		NACT	161	NA	93	NA	76.0	NA	NA
	Willows et al. 2016	RT (a-b: lesion </> 2 cm)	1238 (a)	44	469	NA	66.7	4.5	1.7
			134 (b)	30	10		70.0	11.1	4.2
		Non-radical FSS	124	NA	71	NA	67.6	2.7	0.5
		NACT	62	NA	36	NA	72.2	6.3	1.3
	Nezhat et al. 2020	ST/CKC	283	29.3	131	8	86.4	1.4	0.2
		VRT	1387	43.8	606	78	63.4	3.7	1.1
		AbRT	1060	43	229	104	65.7	3.6	0.7
		RT (MIS)	314	22.6	81	16	56.5	3.3	0.1
	Schuurman et al. 2021	LLETZ/CKC/ST	612	48.5	241	NA	77.4	3.6	0.8
		VRT	1539	55	707	NA	70.6	4.2	1.7
		AbRT (lptm)	1635	48	353	NA	58	3.1	1.5
		RT (MIS)	344	35.7	81	NA	71.6	4.5	1.5
Endometrial cancer	Park et al. 2013	Progestin therapy (PG)	141	49.6	51	44	66.0	31.9	0
	Harrison et al. 2019	PG: alone (1); followed by hysterectomy (2)	421 (1)	NA	131	65	11.6	NA	NA
			397 (2)	NA	34	216	52.0	NA	NA
	Leone et al. 2019	LNG-IUD	44	43.1	14	8	23.4	41.5	NA
	Schuurman et al. 2021	FSS/PG/LNG	505	62.6	256	NA	72	34.7	0.8
Ovarian cancer	Bentivegna et al. 2016	Conservative surgery *	651 (EOC)	42.8	323	NA	69.0	12	6
	Tamauchi et al. 2018	Conservative surgery	105 (OGCT)	42.8	65	7	64.6	10.4	3.8
	Plett et al. 2020	Conservative surgery	95 (BOT)	43.1	48	NA	79.0	13	NA
	Bercow et al. 2021	Conservative surgery	614 (EOC)	50	242	NA	76–96	5–18	8.8
			992 (BOT)	NA	657	NA	23–100	11.6	2.3
	Schuurman et al. 2021	Conservative surgery	750	44.2	280	NA	89.4	15.7	14.7

Note: TTC: trying to conceive; ART: reproductive assisted techniques; LBR: live birth rate; NA: not available; ST: simple trachelectomy VRT—vaginal radical trachelectomy; RT: radical trachelectomy; NACT: neoadjuvant chemotherapy CK: cold knife conization; AbRT: abdominal radical trachelectomy; MIS: minimal invasive surgery (laparoscopic or robotassisted); LLETZ: large loop excision of transformation zone; lptm: laparotomy; EOC: epitelian ovarian cancer; BOT: borderline ovarian tumor; OGCT: ovarian germ cell tumor. *: considered as the preservation of at least one ovary and the uterus.

## Data Availability

All data involved in this study will be made available by the corresponding author upon request.

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
