# Peer review of "Fertility after Cancer: Risks and Successes"

_cancers, 2022, doi:10.3390/cancers14102500_

Round 1

Reviewer 1 Report

I commend the authors on putting together a comprehensive review on this very important and salient topic. The authors have done a comprehensive review of the literature and summarized well.

The review's strong points include the list of cancers. I recommend organizing chronologically, starting from pediatric cancers.

I also would like to request clarification on statistics on cervical cancer. I am not sure what is meant by Nordic countries. I would like to recommend a clarification and re-wording. Nordic is not a commonly used concept nor is it clear what that means.

In general, with all due respect, I'd like to suggest English language review of this valuable manuscript.

Reviewer 2 Report

I have read the manuscript with interest, however, the text needs several serious improvements:

1/ Some more details about the influence of radiotherapy on ovaries should be given

2/ Please give the table with the chemotherapeutic agents and their gonadotoxicity

3/ Please give more details concerning oophoropexy – the sites and possible complications of altered ovarian localization

4/ Table 2 – please add oncological outcomes and give separate results for all kinds of surgery (i.e. vaginal radical trachelectomy, abdominal RT, laparoscopic RT, robotic RT) and the possible factors influencing the differences

5/ Chapter 6.2 – please discuss the possible forms of therapy: PG alone, PG + LNG-IUD, LNG-IUD alone. Please give some details about the control protocols and what are the options in the case of recurrence (another line of conservative treatment or radical surgery?)

6/ Please give more details about MOGCTs and sex-cord tumors therapy and fertility results in adolescents. Give information about Edinburgh OTC criteria (Wallace et al. 2014)

7/ As leukemia and lymphoma are serious problems in adolescents the article should be supplemented with a chapter concerning the problems and fertility outcomes in these kinds of pathology.

Minor remarks:

lines 329, 330, 332 should be GnRH

line 339 ovarian cells I suppose

line 417 should be antiapoptotic

Please check the English language carefully.

Reviewer 3 Report

This study assesses the risks and successes of fertility after cancer and reveals an important health issue properly. It is nice to have the relevant studies collated and discussed all in one place. The manuscript requires major revision.

Lines 45-47 would be advisable to move cervical cancer after the breast cancer to meet the incidence.

Line 66 specifies what type of cancer is found in children and adolescents? (For example, women who survived a childhood neuroblastoma doi: 10.3389/fendo.2019.00555), acute lymphoid leukemia

Line 68 specifies the modalities to preserve prepubertal fertility

Line 259 please detail the effects of radiotherapy on ovarian tissue

  • the oocyte is highly radiosensitive with the LD50of human oocytes estimated at <2 Gy(DOI: 1093/humrep/deg016) and depends on its growth phase, with quiescent primordial follicles being more radioresistant in general relative to larger maturing follicles. 

Line 327 insert the figures’ legend

Line 336 insert references

Line 337 typo corrections “that that”

Line 369 please specify which kind of blood test

Line 417 antipoptotic instead of antiapoptotic

Some recommendations:

  • uterine transplantation for fertility preservation in gynecologic cancers
  • artificial ovary
  • fertility-sparing therapy for vulvar cancer
  • do not use isolated sentences instead of phrases (for example, lines 345-348)
  • present, if possible, the values of sensitivity and specificity of AMH and AFC for different types of cancer

In section 7 there is no textual information for new therapies, and written information should possibly be inserted in section 8.

Kind regards

Round 2

Reviewer 2 Report

The authors have made significant improvements to the text. I have only some minor editorial remarks:

Line 163 should be Van den Berg

References 163 and 164 are incomplete

Lines 190, 198, 199 please remove the dates in the parentheses

Reviewer 3 Report

Dear Authors,   Congratulations on your effort in conducting this review.   One mention of a typo on line 634 - motherhood instead of 16therhood   Kind regards